# Bioactive Compounds in Sea Buckthorn and their Efficacy in Preventing and Treating Metabolic Syndrome

**DOI:** 10.3390/foods12101985

**Published:** 2023-05-13

**Authors:** Ying Chen, Yunfei Cai, Ke Wang, Yousheng Wang

**Affiliations:** 1Key Laboratory of Geriatric Nutrition and Health, Ministry of Education, Beijing Technology and Business University, Beijing 100048, China; chenying@btbu.edu.cn (Y.C.);; 2Rizhao Huawei Institute of Comprehensive Health Industries, Shandong Keepfit Biotech. Co., Ltd., Rizhao 276800, China

**Keywords:** sea buckthorn, metabolic syndrome, bioactive compounds

## Abstract

Sea buckthorn (*Hippophae rhamnoides* L. or *Elaeagnus rhamnoides* L.) is a plant that has long been used as a Chinese herbal medicine. This species is known to contain numerous bioactive components, including polyphenols, fatty acids, vitamins, and phytosterols, which may be responsible for its medicinal value. In experiments both in vitro and in vivo (ranging from cell lines to animal models and human patients), sea buckthorn has shown positive effects on symptoms of metabolic syndrome; evidence suggests that sea buckthorn treatment can decrease blood lipid content, blood pressure, and blood sugar levels, and regulate key metabolites. This article reviews the main bioactive compounds present in sea buckthorn and discusses their efficacy in treating metabolic syndrome. Specifically, we highlight bioactive compounds isolated from distinct sea buckthorn tissues; their effects on abdominal obesity, hypertension, hyperglycemia, and dyslipidemia; and their potential mechanisms of action in clinical applications. This review provides key insight into the benefits of sea buckthorn, promoting future research of this species and expansion of sea buckthorn-based therapies for metabolic syndrome.

## 1. Introduction

Sea buckthorn is a thorny, deciduous, dioecious shrub of the family *Elaeagnaceae*. It is consumed throughout the world for both nutritional and medicinal purposes. Sea buckthorn trees are usually 2–6 m tall, with rough bark and stout young branches that grow from the main trunk in the form of small, sharp spines. It has lance-shaped or linear leaves of 3–8 cm in length and <7 mm in width that are dark green on the adaxial surface and silvery gray on the abaxial surface. It produces yellow or orange spherical berries of 3–8 mm in diameter, which cluster together and are densely surrounded by sharp spines. The ovoid seeds are brown or gray in color, 3–4 mm in length, and covered by a shiny shell (Figure 1). Sea buckthorn fruits are used in popular foods such as bread, yogurt, and jam, and in beverages, such as tea. Sea buckthorn products are a source of bioactive substances, including polyphenols, fatty acids, vitamins, and phytosterols [1].

Sea buckthorn contains nearly 200 known nutrients and bioactive compounds, giving it beneficial nutritional properties [2]. The medicinal value of sea buckthorn was recognized by the Chinese medical system 3000 years ago, dating back to the Tang Dynasty [3]. According to records, in traditional Chinese medicine, sea buckthorn is used to treat diseases including circulatory system diseases, skin lesions, metabolic disorders and digestive system diseases [4]. The value of sea buckthorn in the treatment of gastrointestinal diseases, cardiovascular diseases, and burns was recorded in the Tibetan medicinal classic *rGyud Bzi* (The Four Books of Pharmacopoeia), a classic of Tibetan medicine [5], and was officially listed in the Chinese Pharmacopoeia in 1977 [6,7,8,9]. It reportedly has medical value due to properties that provide anti-oxidative, immune-regulatory, cardioprotective, anti-atherosclerotic, anti-bacterial, anti-viral, anti-inflammatory, anti-diabetic, anti-cancer, hepatoprotective, and skin-protective effects. Sea buckthorn may be a valuable tool in preventing or treating metabolic syndrome. This review summarizes the current knowledge of the bioactive compounds produced by sea buckthorn and their effects in preventing and treating metabolic syndrome, cardiovascular diseases, type 2 diabetes, and other chronic diseases [9,10].

## 2. Bioactive Compounds in Sea Buckthorn

### 2.1. Polyphenols

Polyphenols are the most abundant and widely distributed class of specialized metabolites in plants. They are typically classified based on the number of phenolic rings and other structural elements. All sea buckthorn tissues (namely, the berries, roots, leaves, stems, and branches) contain phenols, but the polyphenol concentrations in the fruits and leaves vary with the specific variety, geographical location, and physiological maturity of a plant. The most abundant polyphenols in sea buckthorn are phenolic acids (Table 1 and Figure 2) and flavonoids (Table 2 and Figure 3) [11,12]. The total phenol content in sea buckthorn (calculated in gallic acid equivalent [GAE]) ranges from 32.93–1417 mg/100 g of tissue [11,13], with variations occurring primarily between sea buckthorn varieties. The free form of gallic acid is the dominant phenolic acid in sea buckthorn. Concentrations differ between the leaves (79 mg/kg) and the berries (16.9 mg/kg). Other phenolic acids, such as caffeic acid, *p*-coumaric acid, and ferulic acid, are present at much lower concentrations in sea buckthorn [14].

Flavonoids are several times more abundant in sea buckthorn berries than in other high-flavonoid plants, such as hawthorn, dogwood cherry, and blueberry [18,19]. Flavonoid glycosides, including isorhamnetin, quercetin, myricetin, and kaempferol, are some of the most abundant phenolic compounds in sea buckthorn [19]. Sea buckthorn seeds contain various flavonoids; glycosides are the most abundant, but there are also small amounts of proanthocyanidins, catechins, triterpene saponins, and some polar and hydrophobic compounds [20]. Drying methods affect the flavonoid concentrations in a final sea buckthorn product [21].

### 2.2. Fatty Acids

Nineteen types of fatty acids have been identified in sea buckthorn, including eight types of saturated fatty acids and 11 types of unsaturated fatty acids. Sea buckthorn contains several essential fatty acids (Table 3 and Figure 4), the most abundant of which are α-linolenic acid and linoleic acid [22,23,24]. In addition, several ω-7 fatty acids, including palmitoleic acid, hexadecatrienoic acid, heptadecenoic acid, and vaccenic acid, are present at high concentrations in sea buckthorn [25]. This plant is also rich in palmitoleic acid, which accounts for 16–54% of the total fatty acid content [26,27]. Therefore, sea buckthorn is of particular interest as a food product because it introduces relatively high levels of palmitoleic acid into the human diet, which cannot be accomplished by the consumption of other foods.

In food products, the most commonly consumed oils are composed of fatty acids with 18-carbon chains and widely varying degrees of unsaturation. However, popular plant fats, such as coconut oil and butter, have fewer than 16 carbons in the main chain, and fatty acids (especially saturated fat) are predominant [28]. Previous studies have shown a positive association between dietary saturated fatty acids content and cardiovascular disease [29,30]. Saturated fatty acids have also been shown to have a cholesterol-raising effect [29]. Excessive intake of long-chain saturated fatty acids can contribute to mitochondrial dysfunction [31], insulin resistance, decreased glycolysis, and endoplasmic reticulum stress [32]. There is also evidence that saturated fatty acids themselves may play a rather limited role in the development of metabolic syndrome [33]. Therefore, the supplementation of sea buckthorn also requires attention to the negative effects of saturated fatty acids on metabolic diseases.

### 2.3. Vitamins

Sea buckthorn plants are rich in vitamins (Table 4) [34]. Most notably, sea buckthorn berries have extremely high vitamin C content [35]; fresh berries average 7950 mg/kg vitamin C [36], which is 12 times higher than the levels found in oranges. The most abundant B vitamins are B_1_ (0.16–0.35 mg/kg fresh weight), B_2_ (0.30–5.0 mg/kg fresh weight) and B_11_ (0–7.9 mg/kg fresh weight) [37]. The primary fat-soluble vitamins in sea buckthorn are vitamins K and E; the latter is a group of eight fat-soluble compounds, namely tocopherols (isomers α, β, γ, and δ) and tocotrienols (isomers α, β, γ, and δ), which have antioxidant activity and are nutritionally essential [38]. α-tocopherol has the highest biological activity of the eight vitamin E compounds [39], and is present in sea buckthorn at levels of 43–223 mg/kg fresh weight [40,41].

### 2.4. Phytosterols

To date, 20 phytosterols have been identified in sea buckthorn [46]. Sea buckthorn contains 20–30 g phytosterols/kg fresh weight, which is 4 to 20 times higher than the levels found in soybean oil. β-sitosterol is the most abundant at ~5.2–5.7 g/kg fresh weight [47,48,49]. The other abundant phytosterols are campesterol, stigmasterol, and ∆^5^-avenasterol (Table 5 and Figure 5) [50].

## 3. Efficacy and Mechanism of Sea Buckthorn Active Compounds in Treating Metabolic Syndrome

Metabolic syndrome is characterized by various metabolic abnormalities, including abdominal obesity, high blood pressure, hyperglycemia, and dyslipidemia. The key aspects of metabolic syndrome treatment include reducing abnormal lipogenesis; improving dyslipidemia and insulin resistance; and controlling blood sugar, blood pressure, and other metabolic pathways. The bioactive compounds in sea buckthorn positively regulate metabolism and ameliorate complications caused by metabolic disorders (Figure 6).

### 3.1. Clinical Trials of Sea Buckthorn

Sea buckthorn has been used in clinical trials to treat metabolic syndrome in recent years (Table 6). A study in obese children assessed the effects of sea buckthorn pulp oil treatment (800 mg/d for 60 d) on inflammation, systemic redox status, and endothelial function. The treatment was shown to prevent atherosclerosis (AS) by strongly reducing triglycerides (TG), cholesterol, and blood pressure, and weakly reducing oxidative stress, inflammation, and insulin resistance [53]. Another human study investigated the effects of dietary supplementation with phenol-rich sea buckthorn powder. An analysis of plasma with nuclear magnetic resonance (NMR) fingerprinting after meals showed that the treatment delays postprandial lipid changes and inhibits increases of 3-hydroxy butanoic acid and *N*-acetyl glycoproteins compared to untreated controls [54].

The effects of sea buckthorn puree on the plasma metabolome and intestinal microflora have been assessed in patients with hypercholesterolemia. After 45 d of treatment, increases in blood glucose, lactate, and lipid levels negated the beneficial effects on sugar and lipid metabolism. This may have been due to the additional sugar intake caused by sea buckthorn puree consumption, which could have negatively affected the metabolism of people with hypercholesterolemia. However, after 90 d, the blood glucose and lactate levels decreased to below the baseline and the blood lipid levels returned to the baseline, potentially due to the actions of the bioactive ingredients (such as the phenolic compounds) in the sea buckthorn puree [56].

In a randomized, controlled, single-blind, three-way crossover study, sea buckthorn treatment was shown to improve blood glucose levels by 44.7% compared to a control group (*p* < 0.01), and to decrease the plasma insulin concentration at 30 and 45 min post-treatment by 39.6% (*p* < 0.01) and 16.5% (*p* < 0.05), respectively. [57]. A separate randomized, double-blind, two-way crossover study showed a slight decrease in fasting blood glucose (FBG) levels in patients with impaired glucose regulation (IGR) after consumption of sea buckthorn puree for 5 weeks [58]. As trans-palmitoleic acid (16:1n-7*t*) is associated with a lower incidence of type 2 diabetes, another study analyzed the effects of unmodified sea buckthorn oil and of a 16:1 sea buckthorn oil–n-7t mixture on serum phospholipid fatty acid (PLFA) levels. Both treatments moderately increased the PLFA concentrations in metabolically healthy adults, suggesting that sea buckthorn oil could prevent diabetes [59]. 

Another study showed that dietary supplementation with 0.75 mL of sea buckthorn seed oil per day effectively reduces dyslipidemia, cardiovascular risk factors, and hypertension in humans, which may be due to the presence of ω-3, -6, and -9 fatty acids in the oil. Improved antioxidant parameters may also be attributed to the high levels of β-carotene and vitamin E [60]. Sea buckthorn puree has also been assessed for its capacity to reduce blood lipid levels and other cardiovascular disease risk factors. It does not affect the serum levels of total cholesterol (TC), low-density lipoprotein cholesterol (LDL-C), or TG, but high-density lipoprotein cholesterol (HDL-C) levels show an overall decreasing trend after increasing at the start of treatment. Diastolic blood pressure (DBP) also decreases after sea buckthorn puree consumption, and hsCRP concentrations show a decreasing trend. These results show that long-term sea buckthorn puree consumption has anti-inflammatory effects and reduces blood pressure in patients with hypercholesterolemia [55]

### 3.2. Hyperlipidemia and Obesity

Lipids play important roles in providing energy and forming essential fatty acids. They are also integral components of human cells and tissues, such as cell membranes and myelin sheaths. Abnormalities in fat metabolism can result in metabolic disorders [61]. Lipid metabolism is also involved in the development of obesity.

Sea buckthorn supplementation has beneficial effects on lipid levels, which can be attributed to the presence of flavonoids (e.g., isorhamnetin, quercetin, and borneol), β-sitosterol, palmitoleic acid, and/or linolenic acid (Table 7). In mice with hypercholesterolemia induced by a high-fat diet (HFD), treatment with sea buckthorn-derived flavone significantly reduces glucose, serum TC, and LDL-C levels in the blood and TC and TG concentrations in the liver [62]. In mice with hypercholesterolemia induced by phenol oxidative stress and a high-cholesterol diet (HCD), treatment with sea buckberry wine decreases oxidized glutathione (GSH) levels and liver lipid peroxidation. This treatment also increases superoxide dismutase (SOD) activity, levels of GSH and lipid peroxidation in the liver, and the ratio of HDL-C to LDL-C [63].

In another study, polyphenols isolated from sea buckthorn berries were orally administered to hyperlipidemic rats at a dose of 7–28 mg/kg. This significantly reduced the blood lipid levels, increased the antioxidant enzyme activity, and decreased the serum levels of tumor necrosis factor-α and interleukin (IL)-6. Furthermore, sea buckthorn berries have been shown to reduce vascular injury in hyperlipidemic rats by decreasing endothelial nitric oxide synthase (eNOS), lectin-like oxLDL receptor-1 (LOX-1), and intercellular adhesion molecule-1 (ICAM-1) expression in the aorta at both the mRNA and protein levels [64].

In HFD-induced obese mice, a 12-week treatment with 0.04% sea buckthorn flavone reversed obesity, liver steatosis, insulin sensitivity, and the inflammatory response. Specifically, the treatment increased energy expenditure and liver fatty acid oxidation (FAO) and inhibited adipose tissue formation, liver adipose formation, and liver fat absorption. Sea buckthorn flavonoids also inhibit plasma gastric inhibitory polypeptide (GIP) levels and liver glycogenase activity, which are regulated by secreted resistin and pro-inflammatory cytokines [69].

Sea buckthorn flavonoids can also regulate peroxide-activated receptor α (PPAR-α) and peroxide-activated receptor γ (PPAR-γ) expression in the liver and adipose tissue in a dose-dependent manner. In mice with HFD-induced obesity, this inhibits adipose tissue inflammation, decreasing obesity and reducing TG levels [70]. In mice fed a high-fat, high-fructose diet (HFFD), sea buckthorn flavonoids alleviate cognitive impairment by effectively normalizing insulin signaling and reducing neuroinflammation [65]. Sea buckthorn flavonoids also improve hyperlipidemia by promoting cholesterol conversion to bile acids and cholesterol effluent, inhibiting de novo cholesterol synthesis, and accelerating FAO [66].

In hypercholesterolemic golden hamsters, treatment with sea buckthorn seed oil (SBSO) downregulates acyl-CoA: cholesterol acyltransferase 2 (ACAT2), microsomal triacylglycerol transporter (MTP), and adenosine triphosphate binding box transporter 8 (ABCG8). SBSO supplementation also increases intestinal short-chain fatty acid production and neutral sterol excretion. A metagenomic analysis showed that dietary supplementation with sea buckthorn seed oil replacing 50% lard (SL) and replacing 100% lard (SH) positively regulates the relative abundance of *Bacteroides S24-7*, *Ruminococcus*, and *Eubacteriaceae* [52].

The hypolipidemic effects of sea buckthorn fruit oil (SBFO) extracts, which are rich in palmitoleic acid, have been studied in mice. Treatment with SBFO extract controls body weight and adipose tissue quality, reduces fat accumulation, and increases TC, TG, HDL-C, and non-HDL-C in a dose-dependent manner. SBFO extract also reduces hyperlipidemia-induced oxidative stress and liver damage by regulating antioxidant enzymes. Quantitative reverse transcription (qRT)-PCR and Western blot analyses indicate that SBFO extract influences expression of key genes in the adenosine monophosphate-activated protein kinase (AMPK) and Akt pathways, and promotes AMPK and Akt phosphorylation [72]. In adipocytes, sea buckthorn oil acts as an insulin sensitizer. Furthermore, it promotes 3T3-L1 preadipocyte proliferation and differentiation; it also increases insulin sensitivity, glucose transporter 4 (GLUT4) expression, and glucose uptake, potentially through AMPK and Akt activation [71].

### 3.3. Hyperglycemia and Diabetes

Diabetes is associated with abnormal carbohydrate, fat, and protein metabolism due to defects in insulin secretion, insulin action, or both. If not properly controlled, diabetes can lead to many complications, such as hyperlipidemia, hypertension, AS, hyperinsulinemia, retinopathy, kidney disease, and peripheral neuropathy [74].

Several studies have evaluated the potential of sea buckthorn supplementation as a treatment for diabetes (Table 8). The results of a study on the antidiabetic activity of sea buckthorn pulp oil on human islet cells showed that sea buckthorn pulp oil enhanced the efficacy of glucose-induced insulin secretion by activating G protein-coupled receptors in pancreatic β-cells. Among the fatty acids of sea buckthorn pulp oil, palmitoleic acid had the highest activity [75]. Zhang et al. investigated the effects of sea buckthorn seed residue water extract on blood glucose and lipid levels and on antioxidant-related parameters in streptozotocin-induced diabetic rats. The rats were divided into four groups: normal control, diabetic control, diabetic treated with 5 mg/kg glyburide, and diabetic treated with 400 mg/kg sea buckthorn seed residue extract. The latter group showed significantly reduced blood glucose, TG, and nitric oxide levels. In addition, the serum SOD activity and GSH levels were significantly increased. This demonstrates the potential hypoglycemic, TG-lowering, and antioxidant effects of sea buckthorn supplements. Furthermore, it suggests that sea buckthorn may prevent some diabetic complications associated with hyperlipidemia and oxidative stress [76].

Sea buckthorn leaf extract has shown strong antioxidant and α-glucosidase-inhibitory activity. Six compounds have been isolated from the leaf extract: kaempferol-3-*O-β-*_D_-(600-*O*-coumaryl) glycoside, 1-feruloyl-*β-*_D_-glucopyranoside, isorhamnetin-3-*O*-glucoside, quercetin 3-*O*-*β*-_D_-glucopyranoside, quercetin 3-*O-β-*_D_-glucopyranosyl-7-*O-α-*_L_-rhamnopyranoside, and isorhamnetin-3-*O*-rutinoside. In a comparison of sea buckthorn leaf extracts generated with a range of polar and nonpolar solvents, butanol leaf extracts were shown to contain the largest number of phenolic compounds, have the highest free radical scavenging activity, and demonstrate the strongest α-glucosidase inhibition [87]. Methanol leaf extracts also positively affect antioxidant and antidiabetic activities in normal and alloxan diabetic Wistar rats in vitro; compared to a diabetic control group, the FBG levels were decreased in alloxan-induced diabetic rats who were intragastrically administered sea buckthorn leaf extract. Furthermore, levels of the endogenous antioxidant enzymes SOD and GSH peroxidase were significantly increased and malondialdehyde levels were significantly decreased in sea buckthorn-treated diabetic rats. These results indicate that a methanol extract of sea buckthorn leaf enhances antioxidant defenses against reactive oxygen species produced as a result of hyperglycemia [79].

L-resveratrol extract (SQE) and L-resveratrol standard (QS) extracted from sea buckthorn leaves show good inhibitory activity against α-amylase. Kinetic studies have shown that the enzyme inhibition is competitive. Glucose consumption by SQE and QS reduce the total TG and non-esterified fatty acid (NEFA) contents in cells with elevated insulin resistance (IR). In addition, SQE and QS downregulate glucose 6-phosphatase and upregulate PPARα. These results suggest that SQE and QS play important roles in regulating glycolipid metabolism [80].

Four unique branched-chain amino acid polypeptides have been identified in sea buckthorn seed proteins: Leu/Ile-Pro-Glu-Asp-Pro, Asp-Leu/Ile-Val-Gly-Glu, Leu/Ile-Pro, and Leu/Ile-Pro-Leu/Ile. In experiments investigating the hypoglycemic activity in db/db mice with type 2 diabetes, oral administration of any of these four branched amino acid peptides significantly reverses symptoms of diabetes and reduces FBG levels via GLUT4 upregulation. The branched-chain amino acid polypeptides also significantly increase muscle glycogen content by downregulating protein kinase B (AKT) and glycogen synthetase 3β (GSK-3β) and increasing glycogen synthetase (GS) activity. In addition, they significantly upregulate phosphatidylinositol 3-kinase (PI3K) at the protein level [82].

Sea buckthorn proteins also exert regulatory effects on intestinal microbes. In mice, treatment with sea buckthorn proteins significantly decreases bodyweight and blood glucose levels and recovers normal levels of *Bifidobacterium*, *Lactobacillus*, *Bacillus*, and *Clostridium globularis*. A metagenomic sequencing analysis has revealed differences in the intestinal microbial community as a result of sea buckthorn treatment. Amplified ribosomal DNA restriction analysis (ARDRA) confirmed that sea buckthorn protein supplementation in mice increases intestinal microbial diversity, as measured by the Shannon (H) and Simpson (E) indices [83].

L-quebrachitol may contribute to the hypoglycemic effects of sea buckthorn. In rats, treatment with sea buckthorn juice reduces food intake, weight gain, random blood glucose levels, and insulin receptor β expression in the liver. Sea buckthorn juice also significantly improves glucose tolerance and pancreatic tissue integrity. Enrichment of sea buckthorn juice with L-quebrachitol alcohol increases fasting plasma insulin levels and influences random blood glucose levels, glucose tolerance, and pancreatic tissue, similar to unenriched sea buckthorn juice [77].

In a study of the hypoglycemic effects of sea buckthorn seed protein (SSP) in streptozotocin (STZ)-induced diabetic ICR mice, SSP showed a significant hypoglycemic effect; compared to the diabetic control mice, the diabetic mice treated with SSP had reduced body weight, FBG, inflammatory factors and insulin (SIN), and lipid content [84]. The effects of sea buckthorn fruit oil extract on type 2 diabetes have also been investigated in vitro using HepG2 cells and diabetic rats. These studies show that sea buckthorn fruit oil extract effectively improves the glucose intake in insulin-resistant cells. Furthermore, it significantly reduces the blood glucose and insulin indices of T2 DM SD rats by regulating the phosphatidylinositol-3-kinase (PI3K)/Akt signaling pathway. Western blot and qRT-PCR analyses showed that sea buckthorn extracts promote PI3K and glycogen synthesis (GS) expression, but inhibit GSK-3β expression [85].

### 3.4. Hypertension and Cardiovascular Disease

Hypertension is related to specific metabolic processes, such as insulin resistance and compensatory hyperinsulinemia. Platelet aggregation, endothelial function, and intracellular free calcium levels ([Ca^2+^] _i_) may influence hypertension development. Improved insulin sensitivity and [Ca^2+^] _i_ could lead to antihypertensive effects [88,89].

There is some evidence that sea buckthorn supplementation can lower blood pressure and prevent cardiovascular diseases (Table 9). Flavonoids are a type of polyphenol that are naturally found in fruits and vegetables, including sea buckthorn. The most abundant flavonoids in sea buckthorn fruits and leaves are isorhamnetin and quercetin, respectively [90]. The antioxidant properties of flavanols indicate that they reduce the risk of cardiovascular disease. Treatment with sea buckthorn total flavonoids protects against myocardial ischemia-reperfusion, tumors, oxidative damage, and aging [91]. Sea buckthorn flavonoids also protect endothelial cells from oxidative LDL-induced damage [92]. In rats susceptible to hypertension, treatment with 0.7 g/kg dried sea buckthorn fruit powder for 60 d improves metabolic processes and alleviates hypertensive stress [93]. Feeding rats a diet high in sucrose significantly increases their systolic blood pressure, plasma insulin and TG levels, and angiotensin II levels in the heart and kidneys. However, supplementation with sea buckthorn seed extract has an antihypertensive effect, which is achieved through blocking of the angiotensin II pathway and improvement of insulin sensitivity [89].

The cardioprotective properties of sea buckthorn oil have been attributed to its high unsaturated fatty acid content [47,97,98]. Sea buckthorn oil protects against myocardial ischemia-reperfusion injury in rats by activating the protein kinase B (Akt)-endothelial nitric oxide synthase (ENOS) signaling pathway. Pretreatment with 20 mg/kg pulp oil stabilizes cardiac function and myocardial GSH levels; it also significantly inhibits lipid peroxidation. Sea buckthorn oil also improves hemodynamics and systolic function, reduces tumor necrosis factor levels, and inhibits lactate dehydrogenase activity, which is a marker of myocardial cell damage [96]. In studies of human subjects with normal lipid levels, sea buckthorn pulp and seed oil exhibit anti-aggregation activity, suggesting that these substances may have beneficial effects on the cardiovascular system as antiplatelet/anti-aggregation factors [99].

Sea buckthorn seed oil has been shown to have significant anti-atherosclerotic properties; one study showed that injection of 1 mL of this oil decreased blood concentrations of TG, TC, LDL-C, and HDL-C. The cardioprotective effects of the oil are likely due to the unsaturated fatty acid, tocopherol, phytosterol, and β-carotene contents; when taken in combination, these compounds may have a synergistic effect on cardiovascular health [99]. In individuals with cardiovascular risk, consumption of sea buckthorn fruit extract significantly reduces TC and LDL-C levels and increases HDL-C levels. However, this effect has not been observed in healthy subjects. The cardioprotective effects of the fruit are likely due to compounds including β-sitosterols and flavonoids [100].

## 4. Conclusions and Future Prospects

In many societies, individuals are increasing their consumption of sugars and fats; these unhealthy eating patterns can lead to metabolic disorders. Most existing drug treatments for metabolic syndrome have adverse effects. Natural bioactive ingredients from plants are therefore an attractive potential alternative to drug therapies.

Sea buckthorn is a plant that has been successfully used in food and medicinal applications and is reported to have various beneficial physiological effects. In this review, we summarized the current progress of sea buckthorn in the treatment of metabolic syndrome, including in vitro and in vivo experiments and clinical trials, and the active substances in sea buckthorn that may be beneficial for the treatment of metabolic syndrome. Sea buckthorn extracts have various therapeutic effects related to metabolic syndrome, including preventing diabetes and related complications, reducing blood pressure, improving abnormal lipid and glucose metabolism, inhibiting glucosidase activity, and protecting the cardiovascular system. This plant and its extracts could contribute to novel approaches to metabolic syndrome treatment, with the primary aim of preventing the development of diabetes, hypertension, and other cardiovascular diseases.

However, most studies have only examined the effects of sea buckthorn on major indicators of metabolic syndrome and have not fully demonstrate the mechanism for its health benefits. Due to the diversity of natural active substances produced by sea buckthorn, the mechanisms of action by which specific the extracts may prevent or treat metabolic syndrome require further research. Sea buckthorn is currently underutilized in food applications and has a great potential to be incorporated into a variety of food formulations. The current research may help to develop functional foods that improve metabolic syndrome. In the future, it will be necessary to investigate the mechanisms of action in-depth for better application of sea buckthorn in food production. On the other hand, the pharmacokinetics, clinical applications, and potential toxic effects of sea buckthorn are also worth exploring. This study provides promising directions for future development of sea buckthorn as a health food or over-the-counter prophylactic drug and of sea buckthorn-based drugs for the clinical treatment of metabolic syndrome.

## Figures and Tables

**Figure 1 foods-12-01985-f001:**
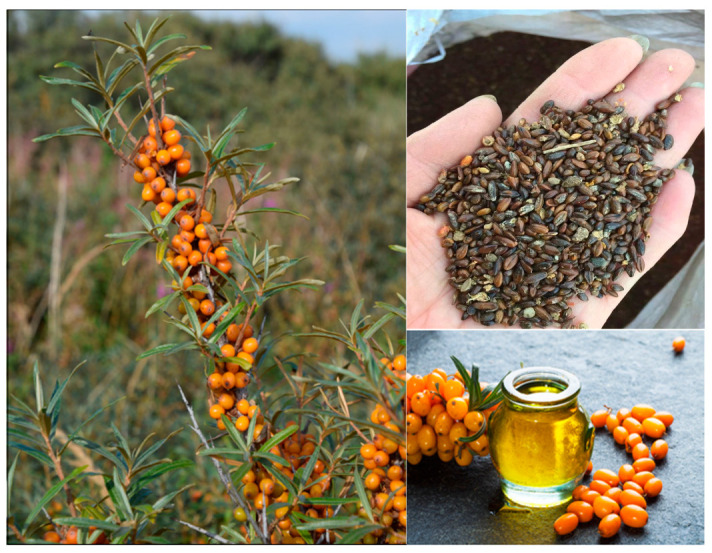
Sea buckthorn berries, leaves, seeds, and oil.

**Figure 2 foods-12-01985-f002:**
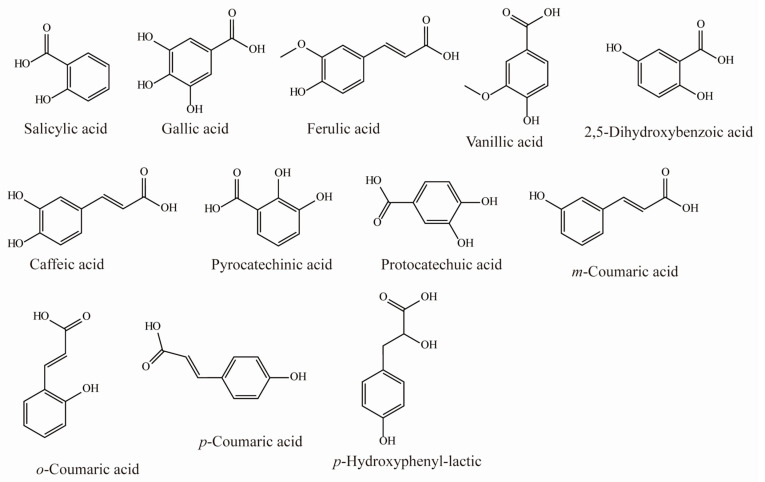
Chemical structures of phenolic acids in sea buckthorn.

**Figure 3 foods-12-01985-f003:**
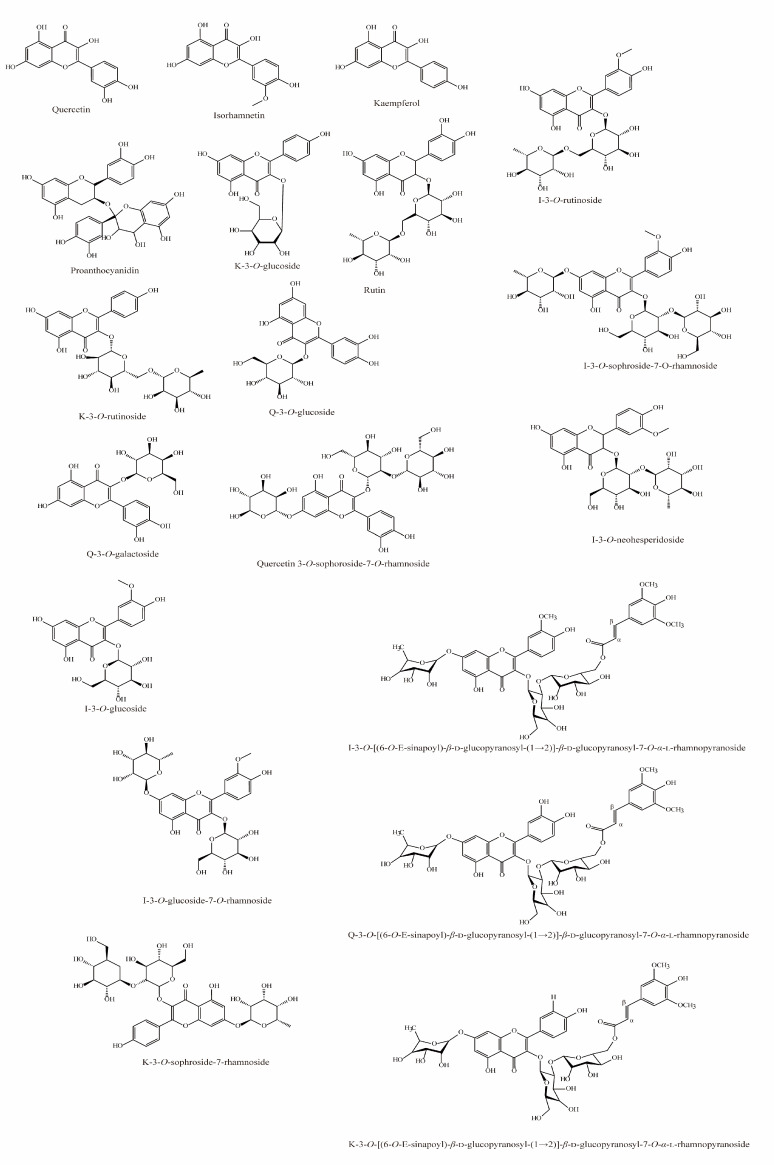
Chemical structures of flavonoids in sea buckthorn.

**Figure 4 foods-12-01985-f004:**
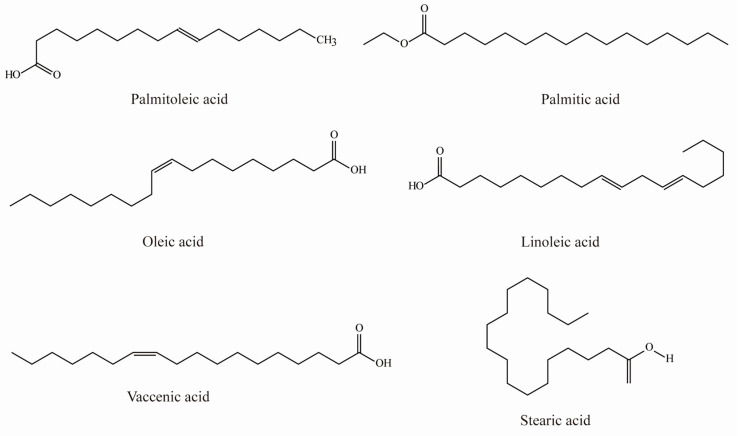
Chemical structures of fatty acids in sea buckthorn.

**Figure 5 foods-12-01985-f005:**
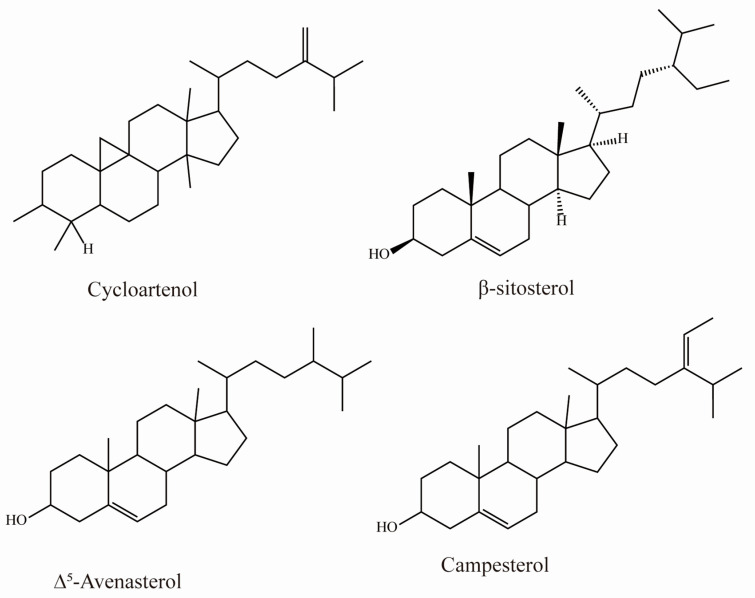
Chemical structures of phytosterols acids in sea buckthorn.

**Figure 6 foods-12-01985-f006:**
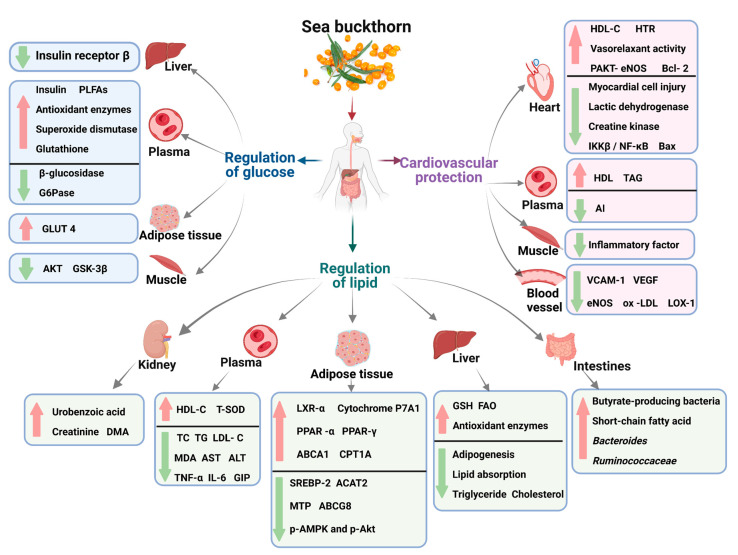
Schematic diagram of the effect of sea buckthorn on metabolic syndrome.

**Table 1 foods-12-01985-t001:** Phenolic acids in sea buckthorn.

Phenolic Acids	Content (mg/kg DM)		Reference
	Berries	Leaves	
2,5-Dihydroxybenzoic acid	0.1–6.1		[15]
Gallic acid	1.0–16.9	19.1–79.1	[14,15]
Pyrocatechinic acid	0.8–6.3		[15]
Protocatechuic acid	0.7–4.3		[15]
Salicylic acid	21–47.5		[15]
Vanillic acid	0.5–1.8		[15]
Caffeic acid	0–6.7	5.9–9.8	[14,15]
*m*-Coumaric acid	0.3–6.1		[15]
*o*-Coumaric acid	2.2–13.3		[15]
*p*-Coumaric acid	1.4–22.3	8.4–13.4	[14,15]
*p*-Hydroxyphenyl-lactic	5.3–24.1		[15]
Ferulic acid	0–10.5	7.2–15.1	[14,15]

**Table 2 foods-12-01985-t002:** Flavonoids in sea buckthorn.

Flavonoids	Content (mg/kg DW)			Reference
	Berries	Leaves	Seeds	
Kaempferol	102	2.8–4.1	60	[11]
Quercetin	40–375	332–1381	110	[11]
Isorhamnetin	103–964		270	[11]
K-3-*O*-glucoside		56–101		[11]
K-3-*O*-rutinoside		291–894		[11]
Q-3-*O*-rutinoside/Rutin	233–288	471–1310	590	[11]
Q-3-*O*-glucoside	402			[11]
Q-3-*O*-galactoside		205–458		[11]
Q-3-*O*-sorphroside-7-*O*-rhamnoside	227–272	301–359	240	[11]
I-3-*O*-rutinoside	210–840	107–496	490	[11]
I-3-*O*-glucoside	260	254–339	280	[11]
K-3-*O*-sophroside-7-rhamnoside	341			[11]
I-3-*O*-sophroside-7-*O*-rhamnoside	308–753	250–446	370	[11]
I-3-*O*-glucoside-7-*O*-rhamnoside	1340	691–1110		[11]
I-3-*O*-neohesperidoside	546–1847	585–1639		[11]
I-3-*O*-[(6-*O*-*E*-sinapoyl)-*β*-_D_-glucopyranosyl-(1→2)]-*β*-_D_-glucopyranosyl-7-*O*-*α*-_L_-rhamnopyranoside	122–300	91–345		[16]
Q-3-*O*-[(6-*O*-*E*-sinapoyl)-*β*-_D_-glucopyranosyl-(1→2)]-*β*-_D_-glucopyranosyl-7-*O*-*α*-_L_-rhamnopyranoside	15–253	751–1821		[16]
K-3-*O*-[(6-*O*-*E*-sinapoyl)-*β*-_D_-glucopyranosyl-(1→2)]-*β*-_D_-glucopyranosyl-7-*O*-*α*-_L_-rhamnopyranoside	75–597	292–1134		[16]
Proanthocyanidins (PA)				
PA Dimers	10–89			[17]
PA Timers	13–95			[17]
PA Tetramers	8–71			[17]
PA oligomers	37–255			[17]
Total PAs	1670–19,410			[17]

**Table 3 foods-12-01985-t003:** Fatty acids in sea buckthorn.

Fatty Acids	Content (% Total Fatty Acid)			Reference
	Berries	Seeds	Pulp/Peel	
Saturated fatty acids	13.70–42.68	10.62–63.92	18.99–71.02	[22]
Monounsaturated fatty acid	40.73–60.37	12.52–22.95	19.66–68.86	[22]
Polyunsaturated fatty acids	3.70–24.62	22.75–76.70	3.55–25.92	[22]
Palmitoleic acid (16:1, x–7)	16–54	0.8–5	15–50	[22,23]
Vaccenic acid (18:1, x–7)	4.55–8.12	2.2	8.1	[22,23]
Palmitic acid (16:0)	17–47	6–11.3	15–40	[22,23]
Oleic acid (18:1, x–9)	2–45.90	15–26	10–26.2	[22,23]
Linoleic acid (18:2, x–6)	3.05–20.13	35–40	4.4–15	[22,23]
Stearic acid		2.6	0.55–1.37	[22,23]

**Table 4 foods-12-01985-t004:** Vitamins in sea buckthorn.

Vitamin	Content (mg/kg FW)					Reference
	Berries	Seeds	Pulp	Seed Oil	Pulp Oil	
α-Tocopherol	43–223	42–160	21–168	444–1550	630–1940	[41,42]
β-Tocopherol	1.3–17	10–16	1.2–19	67–164		[41,42]
γ-Tocopherol	0.50–15	56–95	0.6–13	461–1349	1–3	[41,42]
δ-Tocopherol	0.294			37–113	320–490	[41]
α-Tocotrienol	0.2–4.5					[42]
β-Tocotrienol	0.7–2.3	3–11		30–120		[42]
γ-Tocotrienol	0.3–3.5		0.6–8.4			[42]
δ-Tocotrienol	0.133					[41]
Carotenoids	20–600		300		1200	[1,9]
β-carotene	2–170					[43]
VK	1100–2300					[44]
VC	400–15,500					[36]
VB_1_	0.16–0.35					[45]
VB_2_	0.3–5.0					[45]
VB_11_	0–7.9					[37]

**Table 5 foods-12-01985-t005:** Main phytosterols in sea buckthorn.

Phytosterol	Pulp Oil (g/kg)	Seed Oil (g/kg)	Seeds	Reference
β-sitosterol	3.0–5.7	5.9–13.8	5.6–10.3	[47,51,52]
∆^5^-Avenasterol		3.3	3.5	[51]
Cycloartenol		1.9	1.8	[51]
Stigmasterol	1.0–1.2	0.6–1.2		[52]
Gramisterol		0.6	0.5	[51]
Citrostadienol		0.5		[52]
Campesterol		0.3–1.9	0.3	[51,52]
Total sterols	20–30	10–20		[47]

**Table 6 foods-12-01985-t006:** Clinical trials of sea buckthorn on metabolic syndrome.

Subject	Research Object	Dose	Time	Effects and Mechanisms	Reference
Sea buckthorn pulp oil	A total of 41 overweight children (22 boys and 19 girls) with a mean age of 13.32 ± 4.6 years and 30 healthy children and adolescents (16 boys and 14 girls) with normal weight	800 mg/d	60 d	Lowered inflammation and blood pressure levels, improved plasma lipid profile, and reduced respiratory burst.	[53]
Sea buckthorn berry powder	Fourteen healthy non-smoking males (body mass index 23.5 ± 2.0 kg/m^2^ and age 20–40 years with ca. 26 years)	80 g/d	1 day and 6-day wash-out period	Delayed postprandial increase of lipid levels and the restrained increase of 3-hydroxy butanoic acid and N-acetyl glycoproteins.	[54]
Sea buckthorn puree	A total of 111 patients with hypercholesterolemia	90 g/d	90 d	Lowered the diastolic blood pressure and reduced inflammation biomarker hsCRP.	[55]
Sea Buckthorn Berry Puree	A total of 56 subjects with hypercholesterolemia	90 g/d	90 d	Effectively regulated energy metabolism and intestinal microbiome composition in patients with hypercholesterolemia.	[56]
Sea buckthorn berry	A total of 18 normal-weight subjects	150 g/d	1 day and 2 days apart	Pronounced effect on the postprandial insulin concentration, resulting in a decreased and delayed insulin response.	[57]
Sea buckthorn fruit puree	A total of 38 subjects with IGR	90 mL/day	5 weeks	Downward trend on fasting plasma glucose in subjects with IGR.	[58]
Sea buckthorn Oil	Thirteen participants (7 women and 6 men; age 48 ± 16 y, BMI 30.4 ± 3.7 kg/m^2^)	380, 760, 1520 mg/d	3 weeks, with a 4-week washout phase between the 2 supplements	Increased concentrations of their corresponding phospholipid fatty acids (PLFAs) in metabolically healthy adults.	[59]
Sea buckthorn seed oil	A total of 32 normal and 74 hypertensive and hypercholesterolemic human subjects	0.75 mL/d	30 d	Reduced dyslipidemia and hypertension in male human population.	[60]

**Table 7 foods-12-01985-t007:** The therapeutic effect of sea buckthorn on hyperlipidemia and obesity.

Source	Model	Administration	Active Compounds	Effects and Mechanisms	Reference
Sea buckthorn berries	High-cholesterol diet-fed LACA mice	Oral administration of sea buckthorn wine 20 mL/kg/day for 15 days	Phenolics from sea buckthorn wine	Reduces the TC and LDL-C	[63]
	High-fat diet-fed SD rat	Oral administration of sea buckthorn berries’ polyphenols 7, 14, 28 mg/kg/day for 5 weeks	Polyphenols from sea buckthorn berries	Improves hyperlipidemia tolerance; Prevents aortic endothelial dysfunction; Enhances antioxidant enzyme activity; Decreases the expression level of eNOS, ICAM-1, and LOX-1; Reduces the levels of TNF-α and IL-6.	[64]
	High-fat and -fructose diet-induced C57BL/6J mice.	Oral administration of sea buckthorn flavonoids 0.06%, 0.31% (w/w)/day for 5 weeks	Flavonoids from sea buckthorn	Alleviates insulin resistance, neuroinflammation, and cognitive impairment; Stimulates IRS/AKT activation; Reduces PTP1B expression.	[65]
	Cholesterol-treated HL7702 cells	Incubation with 6.25–100 μM sea buckthorn flavonoids extract	Flavonoids from sea buckthorn	Promotes cholesterol transformation into bile acids and cholesterol efflux; Inhibits cholesterol de novo synthesis; Accelerates fatty acids oxidation; Upregulates the mRNA expression of PPAR-γ, PPAR-α, ABCA1, and CPT1A; Downregulates SREBP-2 and its target gene LDLR; Increases the protein expression of PPAR-γ, LXRα, and CYP7A1.	[66]
	High-fat diet-fed Wistar rat	Oral administration of sea buckthorn powder homogenate 0.5, 2.5, 5 mg/g/day for 28 days Oral administration of sea buckthorn compound stock solution 3.57, 7.14, 14.28 mL/kg/day for 4 weeks	Flavonoids from sea buckthorn powder Flavonoids from sea buckthorn juice	Decreases serum TC, TG, LDL-C, MDA content, AST and ALT activity; Increases serum HDL-C, GSH content, and dT-SOD activity.	[67,68]
Sea buckthorn leaves	High-fat diet-fed C57BL/6J mice	Oral administration of ethanol-extracted sea buckthorn leaves 1.8% (w/w) and flavonoid glycosides from sea buckthorn 0.04% (w/w) leaves /day for 12 weeks	Sea buckthorn leaves flavonoid glycoside	Inhibits adipogenesis in adipose tissue; Inhibits liver fat production and lipid absorption; Improves liver steatosis and promote liver FAO; Reduces plasma GIP levels; inhibits the resistin and proinflammatory cytokines; Improves insulin sensitivity.	[69]
Sea buckthorn seed	High-fat diet-fed ICR mice	Oral administration of ethanol-extracted sea buckthorn seed 50, 100, 150 mg/kg/day for 12 weeks	Total flavonoids from seed residues of *Hippophae rhamnoides* L.	Reduces body, liver, and epididymal fat pad weight in mice; Lowers serum TC and LDL-C levels; Reduces TC and TG concentrations in the liver; Inhibits elevation of blood glucose, improves glucose tolerance abnormalities.	[62]
	High-fat diet-fed C57BL/6J mice	Oral administration of ethanol-extracted sea buckthorn seed 100, 300 mg/kg/day for 9 weeks	Flavonoid-enriched extract from *Hippophae rhamnoides* L. (*Elaeagnaceae*) seed	Suppress PPARγ expression; Upregulates PPARα expression; Decreases TG concentration in serum and liver.	[70]
Sea buckthorn oil	High-fat diet-fed golden Syrian hamsters	Oral administration of sea buckthorn seed oil 50, 100 mg/kg/day for 6 weeks	Fatty acid from sea buckthorn seed oil	Downregulates gene expression of ACAT2, MTP, and ABCG8; Modulates the relative abundance of *Bacteroidales_S24-7_group, Ruminococcaceae, Eubacteriaceae*; Increases intestinal cholesterol excretion.	[52]
	3T3-L1 cells with adipocyte differentiation induction	Incubation with sea buckthorn oil (10^−4^ and 10^−5^*,* volume of SBO/volume of medium(V/V))	Fatty acid from sea buckthorn seed oil	Upregulates proliferating cell nuclear antigen, adipogenic transcriptional factors, mitochondrial biogenesis related gene, glucose transporter 4, greater phosphorylated insulin receptor substrate 1, phosphorylated-Akt, and phosphorylated AMP-activated protein kinase. Promotes 3T3-L1 cells proliferation, adipogenesis, and insulin sensitivity.	[71]
	High-fat diet-fed golden Syrian hamsters	Oral administration of sea buckthorn fruit oil 50, 100, 200 mg/kg/day for 6 weeks	Palmitoleic acid from sea buckthorn fruit oil	Influences the expression of key genes in the AMPK and Akt pathway; Promotes AMPK and Akt protein phosphorylation; Controls body weight and adipose tissue mass; Reduce TC, TG, HDL-C, non-HDL-C level, oxidative stress, liver impairment, and fat accumulation.	[72]
	High-fat diet-fed SD rat	Oral administration of saponification-extracted sea buckthorn 100, 200, 400 mg/kg/day for 42 days	Sterols from sea buckthorn	Decreases TC, TG, LDL-C, Apo-A, HDL-C content, and lipid droplet in cytoplasm; Increases Apo-B, HL, and LPL content; Repairs damaged hepatocyte structures.	[73]

**Table 8 foods-12-01985-t008:** The therapeutic effect of sea buckthorn on hyperglycemia and diabetes.

Source	Model	Administration	Active Compounds	Effects and Mechanisms	Reference
Sea buckthorn berries	db/db mice	Oral administration of sea buckthorn juice 5 mL/kg/day and sea buckthorn juice with 25, 50, 100 mg/kg/day L-quebrachitol for 10 weeks	Flavanol glycosides in sea buckthorn juice	Reduces food intake, weight gain, and blood glucose levels; Decreases expression of insulin receptor β in liver; Improves glucose tolerance and pancreatic tissue integrity.	[77]
	Alloxan-induced diabetic mice	Oral administration of ethanol-extracted sea buckthorn 2, 20, 40 mg/kg/day for 4 weeks	Flavonoids from sea buckthorn	Reduces blood sugar levels and BUN content; Increases insulin level, liver (muscle) glycogen content, and maltase level.	[78]
Sea buckthorn leaves	Alloxan-induced diabetic Wistar rats	Oral administration of methanol-extracted sea buckthorn leaves 200, 400 mg/kg/day for 45 days	sea buckthorn leaf polyphenol	Increases the endogenous antioxidant enzymes, SOD, GSH peroxidase levels; Decreases the malondialdehyde level; Enhances the antioxidant defense against reactive oxygen species.	[79]
	Insulin-resistant HepG2 cells	Incubation with 200 μL sea buckthorn leaf L-resveratrol extract (0.5, 1.0, 2.0, 4.0 mg/mL, diluted with DMEM)	L-resveratrol	Decreases the expression of G6Pase; Increases the expression of PPARα; Inhibits α-amylase activity; Increases glucose consumption; Decreases TG and NEFA.	[80]
	High-sugar diet-fed mice	Oral administration of ethanol-extracted sea buckthorn leaves 240, 480, 960 mg/kg/day for 4 days	*Hippophae rhamnoides* L. *subsp.* *chinensis Rousi* polysaccharide	Enhances the inhibitory effect of α-glucosidase; Improves glucose tolerance; Reduces blood glucose level.	[81]
Sea buckthorn seed	6−8 weeks old db/db mice	Oral administration of purified sea buckthorn seed meal 0.5, 1, 2 g/kg/day for 8 weeks	Hydrophobic branched chain amino acid polypeptide	Upregulates GLUT4 and PI3K protein levels; Downregulates AKT, GSK-3β, and PI3K/Akt protein expression; Increases glycogen synthase activity and muscle glycogen content; Lowers fasting blood sugar levels, weight, and insulin resistance.	[82]
	Streptozotocin-induced diabetic mice	Oral administration of purified sea buckthorn seed 50, 100, 200 mg/kg/day for 4 weeks	Protein from sea buckthorn seed	Reduces body weight and blood glucose level; The *Bifidobacterium, Lactobacillus, Bacteroides, Clostridium coccoides* colony was recovered; Increases gut microbial diversity (H) and Simpson index value (E).	[83]
	Streptozocin-induced diabetic ICR mice	Oral administration of purified sea buckthorn seed 50, 100, 200 mg/kg/d for 4 weeks	Protein from sea buckthorn seed	Reduces CRP, IL-6, TNF-α, and NF-κ B levels; Decreases weight; Decreases FBG; Reduces inflammatory factors and lipid level.	[84]
Sea buckthorn oil	High-fat diet-fed SD rat and glucose-trade HepG2 cells	Incubation with sea buckthorn fruit oil (50, 100, 200, 400 μM, diluted with DMEM) Oral administration of sea buckthorn fruit oil 50, 100, 150 mg/kg/day for 4 weeks	Palmitoleic acid from sea buckthorn seed oil	Promotes the expression of PI3K and GS; Inhibits the expression of GSK-3β; Increases the glucose uptake; Lowers blood glucose; Improves insulin indices.	[85]
	High-fat and -sugar diet-fed SD rats	Oral administration of purify sea buckthorn fruit oil 50, 100, 150 mg/kg/d for 4 weeks	Palmitoleic acid	Activates the PI3K pathway; Promotes GS in skeletal muscle; Relieves insulin resistance; Reduces blood glucose.	[86]
	Glucose-trade human islet EndoC-betaH1 cells	Incubation with sea buck-thorn pulp oil (10μM, diluted with DMEM)	Palmitoleic acid from sea buckthorn pulp oil	Activates G protein-coupled receptors present in pancreatic β-cells; Augments glucose-induced insulin secretion.	[75]

**Table 9 foods-12-01985-t009:** The therapeutic effect of sea buckthorn on hypertension and cardiovascular disease.

Source	Model	Administration	Active Compounds	Effects and Mechanisms	Reference
Sea buckthorn berries	Spontaneously hypertensive stroke-prone rat	Oral administration of powdered *Hippophae* fruit 136 mg/kg/d for 60 days	Polyphenol from powdered *Hippophae* fruits	Decreases mean arterial blood pressure, heart rate, plasma TC, TG, glycosylated hemoglobin, and alkaline phosphatase microvascular capillary partial expression; Reduces hypertensive stress on the ventricular microvessels.	[93]
	High-fat diet-fed SD rat	Oral administration of ethanol-extracted sea buckthorn 7, 14, 28 mg/kg/d for 5 weeks	Flavone from sea buckthorn	Improves SOD activity in liver tissue; Reduces blood lipid level, liver tissue MDA, serum VEGF, VCAM-1 content, aorta eNOS, iNOS protein, and mRNA expression level.	[94]
Sea buckthorn leaves	Endothelial cell line EA hy926 cultured with DMEM	Incubation with sea buckthorn flavonoids	Flavonoids from sea buckthorn	Attenuates ox-LDL-induced LOX-1 upregulation; Increases ox-LDL-reduced eNOS expression; Inhibits superoxide overproduction; Enhances cellular antioxidant defenses.	[92]
Sea buckthorn seed	Sucrose-fed rats	Oral administration of ethanol-extracted sea buckthorn seed residues 50, 100, 150 mg/kg/d for 8 weeks	Total flavones from seed residues of *Hippophae rhamnoides* L.	Suppresses elevated hypertension, hyperinsulinemia, and dyslipidemia.	[89]
	High cholesterol diet-fed white albino rabbits	Oral administration of supercritical-extracted sea buckthorn seed oil 1 mL/d for 30 days	Fatty acids from sea buckthorn seed oil	Reduces plasma cholesterol, LDL-C, AI, and HDL-C levels; Reduces LDL/HDL ratio and HDL-C/TC ratio; Increases vasorelaxant activity of the aorta.	[95]
Sea buckthorn oil	IR-induced Wistar rats	Oral administration of sea buckthorn pulp oil 5, 10, 20 mL/kg/d for 31 days	Fatty acids from sea buckthorn seed oil	Increases expression of Akt–eNOS and Bcl-2; Decreases expression of IKKβ/NF-κB and Bax; Inhibits lipid peroxidation; Reduces malondialdehyde levels, tumor necrosis factor and activities of myocyte injury marker enzymes, lactate dehydrogenase, and creatine kinase-MB; Improves hemodynamic and contractile function.	[96]

## Data Availability

All relevant data are included in the article.

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
