# Peer review of "Bioactive Compounds in Sea Buckthorn and their Efficacy in Preventing and Treating Metabolic Syndrome"

_foods, 2023, doi:10.3390/foods12101985_

Round 1
Reviewer 1 Report
Dietary interventions that promote lifestyle changes are welcomed as an appropriate alternative to reduce metabolic syndrome. Sea buckthorn is an important plant because of its immense medicinal and therapeutic potential. Although numerous reviews on sea buckthorn have been published, it’s always very welcome to draw new respective conclusions and/or discuss its relation to various conditions in health and disease. However, to become a helpful article, a selected aspect that the authors mentioned in their manuscript must be discussed in a broader way to make sense.
1. In order to distinguish this article from many others, I suggest that the authors also present the controversies that may be associated with the use of sea buckthorn in the context of metabolic diseases. Here I would emphasize certain ingredients such as saturated acids that may have a deleterious effect.
2. I ask the authors to prepare a table compiling clinical trials concerning the use of sea buckthorn
3. Not all relevant references are cited (e.g., https://www.sciencedirect.com/science/article/pii/S2352364622000104). Please review the literature more thoroughly
4. Typos, e.g., Latin names in italics (family Elaeagnaceae)
Reviewer 2 Report
This review highlights the beneficial effects of sea buckthorn components on metabolic syndrome. The article is well organized. However, there are several points that could be improved.
In the article, some sentences such as “ancient plant”, “high medicinal value” should be toned down. For example, all plants are ancient.
In Figure 1, at plasma, there are presented effects that correspond to muscle and adipose tissue. Both tissues are missing in the figure.
An additional figure is needed. In this figure, the signaling pathways involved in the effects of the plant should be indicated.
Reviewer 3 Report
Dear Editor and Authors,
The manuscript ‘Bioactive compounds in sea buckthorn and their efficacy in preventing and treating metabolic syndrome’ by Ying Chen 1 , Yunfei Cai 1 , Ke Wang 1 and Yousheng Wang is a review on sea buckthorn composition and potential influence on health especially on reducing the occurrence of or even treating the metabolic syndrome. The manuscript is interesting, well written and the 86 references are a decent collection of publication on the topic:
· Line 244 repetition ‘butanol leaf extracts’
· Line 252 delete ‘are’
· Line 349
· For the first time beta-carotene in sea buckthorn is mentioned. Beta-carotene should be added as a vitamin content in Table 4.
That is why I recommend minor revision,
Yours sincerely
Reviewer 4 Report
In this review, the authors summarize the biological effects of the main bioactive compounds of sea buckthorn, in particular on the metabolic syndrome. The article is well written, but there are a number of concerns that need further clarification.
My comments are as follows.
1. There are a number of excellent reviews in this area from other groups (also cited by the authors) that have been published recently. In particular: Wang Z et al. (2022) Phytochemistry, health benefits, and food applications of 403 sea buckthorn (Hippophae rhamnoides L.): A comprehensive review. Front Nutr 404 9:1036295.
Authors should mention what is new in their paper, what aspects were not covered in the other reviews and are discussed in this article, etc.
2. The illustrations are very sparse. At the very least, the authors should include a photo of the plant, including the berries, perhaps of the seeds. This would help to attract and increase the interest of the readers in this article. Perhaps the authors could also add a paragraph on the history of the use of this plant in traditional Chinese medicine.
3. Structural formulas of the main compounds described in this document are not shown. The authors should show the structural formulas of the most important compounds in this plant, even if this has been done previously (e.g. in the paper of Wang Z et al. 2022).
4. Table 1. “o”-, “p”-, “m”- (ortho, meta and para), e.g., o-, m- and p-coumaric acid should be written in italics; also Table 2: “O”, e.g. in K-3-O-glucoside; etc.
5. Line 79: Ω-7 fatty acids. Please use the lower-case Greek letter. The same in line 357.
6. Line 105 and Table 4: a-, b-, c-, d-tocopherol. Please use the Greek letters instead of a, b, c, d.
7. Line 221: “2010” can be omitted.
8. Lines 311 and 314: “[Ca2+] i”. “i” should be subscript.
9. At the end of their paper, the authors should include a paragraph or a chapter on the tasks of future research and describe which scientific questions are still open or have not yet been sufficiently investigated.
10. The reference should be written consistently. Sometimes the full titles of the journals are written, sometimes the abbreviations; sometimes all words in the titles start with capital letters, sometimes not, or only capital letters are used (e.g. ref. 53).
11. There are a number of typographical errors that should be corrected.
I think it is important that the authors make it clear that their article contains new information that is essential and has not been extensively discussed in previously published reviews.
Round 2
Reviewer 1 Report
The authors improved the content as requested.
Reviewer 4 Report
The quality of this manuscript has now been considerably improved.
All questions have been answered.